# AI-Based Chest CT Analysis for Rapid COVID-19 Diagnosis and Prognosis: A Practical Tool to Flag High-Risk Patients and Lower Healthcare Costs

**DOI:** 10.3390/diagnostics12071608

**Published:** 2022-07-01

**Authors:** Giovanni Esposito, Benoit Ernst, Monique Henket, Marie Winandy, Avishek Chatterjee, Simon Van Eyndhoven, Jelle Praet, Dirk Smeets, Paul Meunier, Renaud Louis, Philippe Kolh, Julien Guiot

**Affiliations:** 1Icometrix, 3012 Leuven, Belgium; giovanni.esposito@icometrix.com (G.E.); avi.chatterjee.ds@gmail.com (A.C.); simon.vaneyndhoven@icometrix.com (S.V.E.); jelle.praet@icometrix.com (J.P.); dirk.smeets@icometrix.com (D.S.); 2Pneumology Department, University of Liège, 4000 Liège, Belgium; benoit.ernst@uliege.be; 3Respiratory Medicine Department, University Hospital of Liège, 4000 Liège, Belgium; monique.henket@chuliege.be (M.H.); marie-laurence.winandy@chuliege.be (M.W.); r.louis@chuliege.be (R.L.); 4Radiology Department, University Hospital of Liège, 4000 Liège, Belgium; paul.meunier@chuliege.be; 5Information Systems Management Department, University Hospital of Liège, 4000 Liège, Belgium; philippe.kolh@chuliege.be

**Keywords:** COVID, SARS-CoV-2, AI-based CT-scan analysis, hospital days reduction, infection reduction, patient flow management, PCR test rationalization, incremental cost-effectiveness ratio, COVID-19 infection spread prevention

## Abstract

Early diagnosis of COVID-19 is required to provide the best treatment to our patients, to prevent the epidemic from spreading in the community, and to reduce costs associated with the aggravation of the disease. We developed a decision tree model to evaluate the impact of using an artificial intelligence-based chest computed tomography (CT) analysis software (icolung, icometrix) to analyze CT scans for the detection and prognosis of COVID-19 cases. The model compared routine practice where patients receiving a chest CT scan were not screened for COVID-19, with a scenario where icolung was introduced to enable COVID-19 diagnosis. The primary outcome was to evaluate the impact of icolung on the transmission of COVID-19 infection, and the secondary outcome was the in-hospital length of stay. Using EUR 20000 as a willingness-to-pay threshold, icolung is cost-effective in reducing the risk of transmission, with a low prevalence of COVID-19 infections. Concerning the hospitalization cost, icolung is cost-effective at a higher value of COVID-19 prevalence and risk of hospitalization. This model provides a framework for the evaluation of AI-based tools for the early detection of COVID-19 cases. It allows for making decisions regarding their implementation in routine practice, considering both costs and effects.

## 1. Introduction

Early identification of patients infected with COVID-19 remains a priority for several reasons, regardless of the severity of their symptoms. In-hospital management of patient flow, whether for consultations or stays, is a critical issue in a pandemic situation, as many people arriving at the hospital may be healthy carriers, hence, making identification complex.

The cross-infection question is of utmost importance in the matter of hospital hygiene. Indeed, patients in contact with each other are at high risk of contamination, often due to insufficient individual protection equipment. Considering the fragility of patients, justifying their presence in the hospital, it is important to implement strategies that lower the cross-infection risks within the hospital as much as possible.

Computed tomography (CT) imaging was widely used during the COVID-19 pandemic [1]. CT findings associated with COVID-19 include bilateral pulmonary parenchymal ground-glass and consolidative pulmonary opacities [2], sometimes with a rounded morphology and a peripheral lung distribution [3]. Mild to moderate disease progression is manifested by an increase in the extent and density of lung opacities [4]. Several systems standardizing the assessment and reporting of COVID-19 on non-enhanced chest CT were proposed [5,6,7].

In addition, some viral infections cause symptoms similar to COVID-19, which may lead to doubts about the diagnosis [8]. People with a high COVID-19 viral load should benefit from a differential diagnosis. The use of immunotherapies, neoplasia, and chronic obstructive pulmonary disease can also be a cause for suspicious radiological signs that can mimic COVID-19 infection in high-risk patients [9,10].

Recently, algorithms based on artificial intelligence (AI) allowed for the rapid discrimination between COVID-19 and other types of pneumonia [11] These tools allow for the rapid detection of infected patients, as well as cross-infections. Automated diagnosis of COVID-19 from CT imaging greatly assisted clinical decision-making, through enhanced reproducibility and diagnostic accuracy, along with speeding up the image interpretation.

Numerous applications of AI, in different forms, are proposed to facilitate several clinical tasks in the management of COVID-19. AI-based tools can be trained to either estimate the diagnosis (e.g., [12,13,14]) or prognosis (e.g., [11,15]) directly from the CT image (or from extracted radiomic features (e.g., [16]), or to perform the ‘intermediate’ task of automatically segmenting/delineating the lungs and/or lung abnormalities in the image (e.g., [17,18]).

One such tool is icolung (icometrix), which performs an AI-based chest CT analysis for the detection and prognosis of COVID-19 infection from chest CT scans. The software quantifies clinically relevant parameters from a non-contrast CT scan, which are summarized in a structured report, along with an early warning score that can guide clinical decisions and management. Therefore, icolung could improve COVID-19 diagnosis and prognosis in patients who receive chest imaging, thus, potentially reducing the risk of spreading, and of severe complications due to the disease.

To better understand the value of icolung in the clinical workflow, we developed a model to evaluate the impact of using icolung for the detection and prognosis of COVID-19 cases in patients receiving routine CT scans in a university hospital setting in Belgium.

As endpoints, we focused on the transmission of COVID-19 infection, expressed as cost per avoided infection (primary outcome), and the in-hospital length of stay (LOS) of COVID-19 patients, expressed as cost per avoided hospital days (secondary outcome). 

This model provides a framework for the evaluation of AI-based tools for the early detection of COVID-19 cases from chest CT scans, and it allows for decisions regarding their implementation in routine practice, considering both costs and effects. 

## 2. Materials and Methods

### 2.1. Icolung Software

The software uses deep learning models that sequentially perform fully automated segmentation of the lungs, lung lobes, and lung abnormalities (ground-glass opacity and consolidation). These convolutional neural network models were based on the 2D and 3D U-net architectures described in [19,20], and were trained, validated, and tested on clinical CT scans, along with voxel-level delineations of lung abnormalities, created by radiological experts.

Based on the models’ predicted masks for lung abnormalities and lobes, the lung involvement in each lobe is computed as the ratio of abnormality volume vs lobe volume, and from that, a lobe-specific severity score (0–5) is derived [21]. The five severity scores are then summed into a global severity score (0–25) for the patient’s current CT exam [21].

The software produces a pdf report with the following content:-The 3D segmentation masks of the abnormalities are visualized in 2D axial and coronal views on a report;-A table with the lung involvement percentages and the corresponding severity scores of both abnormality types. These values are shown for each lung lobe, as well as for the total lungs.

### 2.2. Model Structure and Parameters

We constructed a decision tree analytical model, in which we compared a routine practice (RP) scenario where patients receiving a CT scan in the hospital were not screened for COVID-19 with a scenario where icolung was introduced into the RP to analyze CT scans for the detection and prognosis of COVID-19 cases.

The primary outcome of the model was the cost per avoided infection, and the secondary outcome was the avoided cost per hospital day. Figure 1 shows the decision analytic model structure.

The model assumed that in the routine practice, patients with COVID-19 were undetected and sent home. Some of them developed symptoms and some required hospitalization, which can lead to a short stay, a long stay, or intensive care unit (ICU) admission, depending on the severity. In the alternative scenario, icolung was used for the detection of COVID-19 patients and the early detection of the more severe cases. In the icolung scenario, a polymerase chain reaction (PCR) test was always used to confirm the positive cases.

To estimate the model input parameters, we used published data from the literature, as well as unpublished data, and assumptions based on expert opinion.

We estimated the prevalence of COVID-19 in the community by using the biweekly diagnosed cases from the Johns Hopkins Coronavirus Resource Center, as well as the estimated population of Belgium. We assumed that omicron and delta variants accounted for all the COVID-19 infections, as suggested by our epidemiological data from hospital cases.

We used hospitalization and ICU rates of the two variants to derive the proportion of patients who will be hospitalized, as well as their prognosis in terms of short/long stay and ICU. We used the reproduction number to estimate the number of cases directly generated by one infected individual. We estimated separated reproduction numbers for the community and the healthcare setting as described in [22]. The risk of transmission was adjusted considering hospital protective measures, and both self-isolation and household quarantine.

The model assumed that all detected cases resulted in self-isolation and household quarantine. It also assumed that hospital protective measures were used for all the hospitalized cases in the two scenarios.

Unit hospitalization/ICU costs per patient per day were used to estimate the hospital cost for each scenario, depending on LOS. PCR testing costs were applied in both scenarios for the hospitalized cases. When there was no agreement between icolung and PCR tests, the model assumed that the PCR test was repeated, and gave the correct outcome (meaning the one corresponding with the patient disease status). In these cases, the cost of the PCR test was counted twice. Finally, we assumed that the use of icolung would have a positive impact on the prognosis of the detected cases, reducing by 10% the risk of long hospital stays and ICU admission, as indicated by our preliminary study.

The primary outcome was the cost to prevent one COVID-19 infection in the community, and the secondary outcome was the cost to avoid one hospital day. Primary and second outcomes were measured as incremental cost-effectiveness ratio (ICER), by dividing the cost difference (icolung scenario—RP) by the number of avoided infections, or avoided hospital days. For the primary outcome, we assumed a cost-effectiveness (willingness-to-pay) threshold of EUR 20,000, which reflected the immediate costs of a COVID-19 infection [23].

In addition to the base case analysis, we performed a one-way sensitivity analysis, which allowed us to identify the parameters with the highest impact on the model outcomes. Each parameter in the analysis was varied between its lower and upper 95% confidence or credible interval, or by 50% of its mean value if statistical measures of variance were not available. Finally, we performed two-way sensitivity to assess the impact of varying critical parameters on icolung’s cost-effectiveness. The model was developed using Microsoft Excel 2021 (Microsoft^®^), and is provided in Appendix A. Table 1 summarizes the model input parameters and the range values used for the sensitivity analysis.

## 3. Results

### 3.1. Evaluation of Costs Avoided Using Icolung

In the base case assuming a COVID-19 prevalence of 0.4%, we evaluated that icolung would prevent about 18 infections per 1000 patients, while the impact of icolung on the number of avoided hospital days is 0.1 day per 1000 patients, as shown in Table 2. 

The marginal impact on the number of avoided hospital days is probably because of the low risk of hospitalization assumed in the base case.

The resulting costs to avoid one infection and one hospital day (ICERs) are EUR 8.221 and EUR 2.047.902, respectively (Table 3).

Therefore, for the primary outcome, icolung is cost-effective at the assumed threshold of EUR 20,000. For the secondary outcome, icolung is not cost-effective, given the low rate of hospitalization in the base case analysis. However, it could be cost-effective under certain circumstances (see below).

### 3.2. Sensitivity Analysis

In sensitivity analysis, for the primary outcome (avoided infection), the model is most sensitive to the prevalence of COVID-19, reproduction number, and, to a lesser extent, to the PCR test cost and risk reduction of self-isolation and household quarantine (Figure 2). 

For the secondary outcome (avoided hospital days), the model is most sensitive to the prevalence of COVID-19, the prevalence of the omicron (and, indirectly, the delta) variant, the hospitalization rate of the delta variant, the cost of the PCR test, and the risk reduction in both long stay and ICU admission (Figure 3).

Thus, icolung cost-effectiveness might change, depending on the risk of hospitalization.

In two-way sensitivity analyses, we further examine the impact of the model drivers on the cost-effectiveness of icolung. First, we simultaneously vary the prevalence of COVID-19 and the reproduction number (Rt), to reflect reported estimates in different phases of the pandemic. For the number of avoided infections, icolung is cost-effective (ICER below EUR 20,000), with an already low prevalence of COVID-19 infections and low Rt values (Table 4).

Second, we evaluate further icolung cost-effectiveness in reducing hospital days by varying COVID-19 prevalence and risk of hospitalization, to reflect estimates in different phases of the pandemic, as well as the presence of more severe variants, or patient subgroups with a higher risk of hospitalization. It is proven that icolung is more cost-effective in a higher value of COVID-19 prevalence and risk of hospitalization (Table 5).

## 4. Discussion

In this analysis, we explored the potential health–economic impact of using an AI-based chest CT analysis software (icolung, icometrix) for the detection and prognosis of COVID-19 cases in patients receiving a CT scan in a hospital setting in Belgium. We evaluated the impact of the technology on preventing further spreading of the infection in the community (societal perspective), and on reducing the impact on the hospital resource (health care perspective).

From a societal perspective, this analysis indicates that adding icolung to RP for screening patients receiving chest CT is a cost-effective strategy for preventing infections in the community.

Already at relatively low disease prevalence (>4%) and low circulation (Rt = 0.52), icolung is cost-effective, and even more effective in higher disease prevalence and circulation.

Our model is sensitive to the amount we are willing to pay to prevent a COVID-19 infection in the community. The chosen threshold of EUR 20,000 is a rough estimate, based on direct costs and lost wages. This amount may vary depending on the location and wealth of the community.

From a health care perspective, we also evaluated the impact of icolung technology on reducing the length of hospitalization. Although we did not set a threshold amount of willingness to pay for this, the analysis suggests that this technology is not cost-effective, particularly when disease prevalence is low and hospitalization risk is low. However, when disease prevalence (>30%) and risk of hospitalization (>6%) are high, icolung use becomes more cost-effective. It may, therefore, be beneficial to use it in certain circumstances, such as at the peak of the epidemic in a given area, or in the presence of more virulent variants, or in a group of patients with a high risk of hospitalization.

Other authors used AI analysis on chest CT scans either to improve diagnosis [31,32]), or to predict evolution of diseases lesions [33,34]. Another non-invasive and readily available predictive tool for poor prognosis is the electrocardiogram (ECG) and its 7-day evolution [35,36]. As cardiovascular diseases were identified as a risk condition during a COVID-19 infection, preliminary studies show the benefit of administering low-molecular-weight heparin, in terms of reducing in-hospital mortality [37].

## 5. Strengths and Limitations of This Study

A strength of this study is the ability to evaluate large ranges of probabilities, given the currently limited data and the dynamic evolution of the pandemic. Additionally, this is the only decision analysis we are aware of that evaluated an AI-based tool for the detection of COVID-19 cases from chest CT scans. Our model can be extended to other public health contexts and countries, by varying basic parameters such as prevalence, virus reproduction rate, cost per hospitalization day, etc.

The main limitation is the uncertainty around the data that determine the model parameters, and a model is highly dependent on the quality of the data on which it is based. In addition, as the COVID-19 pandemic evolves daily (variants, prevalence, virus reproduction rates), the information on which this analysis is based can quickly become outdated. To account for this limitation, we varied the estimated parameters of our baseline scenario widely. Additionally, we did not consider the long-term consequences of COVID-19 infection, because these consequences are not fully known and difficult to quantify at this time. However, once available, updated costs regarding the long-term consequences of COVID-19 infection could be included in a future model. Finally, the model only considers “primary” transmission of the disease from infected patients, and does not consider “secondary” infections, which would require different and more complex analytical techniques.

Another limitation of this study comes from it being a single-center and retrospective study. Further studies are needed to validate the model with other datasets.

## 6. Conclusions

Overall, we show that this model could provide a different approach to the current pandemic. It allows us to make decisions on hospital policy and resource allocation, considering both costs and effects.

## Figures and Tables

**Figure 1 diagnostics-12-01608-f001:**
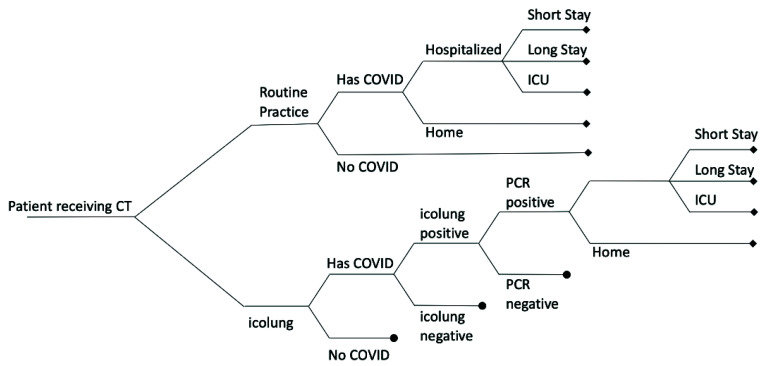
Decision analytic model. All branches terminating in a circle are collapsed to facilitate display, and are the same as branches already open.

**Figure 2 diagnostics-12-01608-f002:**
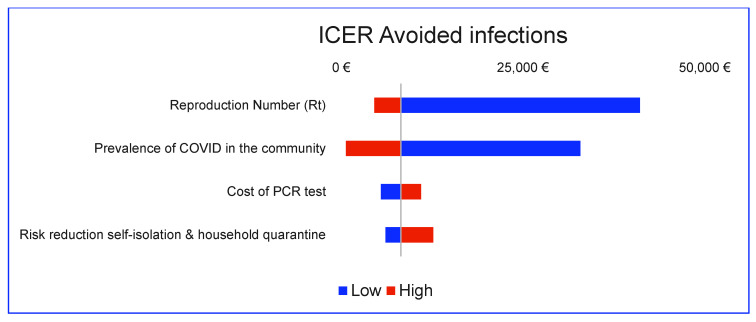
Tornado diagram: primary outcome, RP vs. RP with icolung.

**Figure 3 diagnostics-12-01608-f003:**
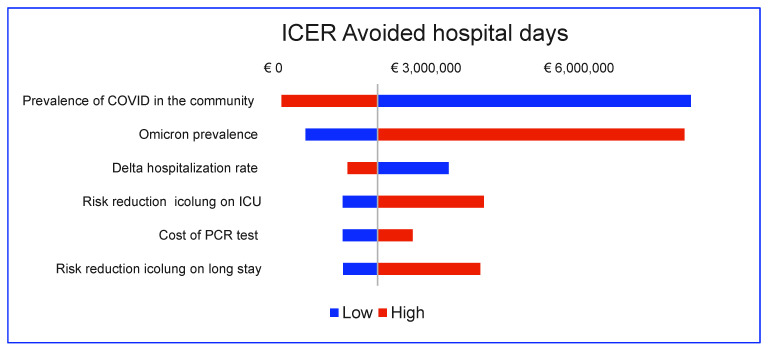
Tornado diagram: secondary outcome, RP vs. RP with icolung.

**Table 1 diagnostics-12-01608-t001:** Parameters and range used in our model for the sensitivity analysis.

Variable	Base Case Value	Range Considered in the Sensitivity Analysis	Reference
Prevalence of COVID in the community	4.00%	1.00–50.00%	[24]
Omicron prevalence	75.00%	0.00–100.00%	Unpublished data
Delta prevalence	25.00%	-	Unpublished data
Omicron hospitalization rate	0.20%	0.10–0.30%	[25]
Delta hospitalization rate	1.10%	0.55–1.65%	
Probability of hospitalization	7.70%	-	Estimated *
Omicron ICU rate (among hospitalized)	24.00%	3.85–11.50%	[25]
Delta ICU rate (among hospitalized)	0.43%	12.50–36.00%	
Probability of ICU admission (among hospitalized)	17.65%	-	Estimated **
Probability of short stay (1.5 days)	18.25%	-	Estimated ***
Probability of long stay (5 days)	64.11%	-	Estimated ****
Sensitivity PCR test	96.20%	91.00–98.40%	[26]
Specificity PCR test	98.70%	95.00–99.00%	
Sensitivity icolung	96.00%	94.00–99.00%	[27]
Specificity icolung	60.00%	59.00–61.00%	
Cost of hospitalization per patient per day	EUR 1000.00	EUR 500.00–1500.00	Assumption
Cost of ICU per patient per day	EUR 3000.00	EUR 1500.00–4500.00	Assumption
Cost of PCR test	EUR 100.00	EUR 50.00–150.00	Assumption
Cost of CT chest scan	EUR 300.00	EUR 150.00–450.00	Assumption
Cost of icolung per patient	EUR 50.00	EUR 25.00–75.00	Assumption
Average hospital short stay duration (days)	1.50	-	Expert opinion
Average hospital long stay duration (days)	5.00	-	Expert opinion
Average ICU stay duration (days)	14.00	-	Expert opinion
Risk reduction icolung on long stay	0.90	0.80–1.00	Expert opinion
Risk reduction icolung on ICU	0.90	0.80–1.00	Expert opinion
Reproduction number (community)	1.25	0.25–2.25	
Reproduction number short stay	1.25	-	Assumed to be as community
Reproduction number long stay	1.87	-	Estimated *****
Reproduction number ICU	2.19	-	
Risk reduction self-isolation plus household quarantine	0.63	0.50–0.76	[28]
Risk reduction personal protection equipment	0.07	0.06–0.08	[29]

* Estimated as the weighted average of omicron and delta hospitalization rates. ** Estimated as the fraction of ICU patients (omicron + delta) among the hospitalized. *** Based on length of stay distribution data (unpublished): median values 1.5 days (omicron), 5 days (delta). The proportion of patients with a short stay is estimated as 50% of the omicron hospitalized. **** Proportion of patients with long stays is estimated as total hospitalized minus patients with short stays and admitted to ICU. ***** Estimated from the reproduction number for the community as described in [22], adjusting cumulative minute of contact per day according to the type of care [30].

**Table 2 diagnostics-12-01608-t002:** Base case analysis results: costs and outcomes per 1000 patients.

Strategy	Estimated Costs (EUR)	Incremental Costs (EUR)	Infections	Hospital Days	Infections Avoided	Hospital Days Avoided
**Routine practice (RP)**	301,910		49.81	1.02		
**RP + icolung**	453,129	151,220	31.41	0.95	18.4	0.07

**Table 3 diagnostics-12-01608-t003:** Base case analysis results, incremental cost-effectiveness ratio (ICER): costs to avoid one infection and one hospital day.

Outcomes	ICER
**Infections avoided**	EUR 8221
**Hospital days avoided**	EUR 2,047,902

**Table 4 diagnostics-12-01608-t004:** Two-way sensitivity analysis primary outcome (infections avoided).

Prevalence COVID	Rt	ICER
1.0%	0.250	EUR64,480
4.0%	0.520	EUR 19,761
11.0%	0.790	EUR 4726
16.0%	1.060	EUR 2420
21.0%	1.330	EUR 1468
26.0%	1.600	EUR 985
31.0%	1.870	EUR 707
36.0%	2.140	EUR 531
41.0%	2.410	EUR 414
46.0%	2.680	EUR 332

**Table 5 diagnostics-12-01608-t005:** Two-way sensitivity analysis secondary outcome (reducing hospital days).

Prevalence COVID	Hospitalization Risk	ICER
**1.0%**	0.4%	EUR 8,585,600
**6.0%**	1.3%	EUR 525,664
**11.0%**	2.1%	EUR 182,785
**16.0%**	3.3%	EUR 82,140
**21.0%**	4.3%	EUR 47,747
**26.0%**	5.3%	EUR 30,949
**31.0%**	6.3%	EUR 21,508
**36.0%**	7.3%	EUR 15,680
**41.0%**	8.3%	EUR 11,831
**46.0%**	9.3%	EUR 9157

## Data Availability

The dataset generated and/or analyzed during the current study is not publicly available, because these data are considered sensitive, but are available from the corresponding author on reasonable request.

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
