# Peer review of "AI-Based Chest CT Analysis for Rapid COVID-19 Diagnosis and Prognosis: A Practical Tool to Flag High-Risk Patients and Lower Healthcare Costs"

_diagnostics, 2022, doi:10.3390/diagnostics12071608_

Round 1

Reviewer 1 Report

The topic is really interesting and authors have written the manuscript in a well-directed manner. However, here are my comments to increase the acceptability towards a broader readership. 

1. I recommend changing the title to this one: " AI-based chest CT analysis for rapid COVID-19 diagnosis and prognosis: a practical tool to flag high-risk patients and lower healthcare costs.

2. In the introduction, please highlight the AI-based applications during the COVID-19 period, please write a section on the imaging and analysis part modulated by AI. 

3. There are many parts of AI, such as machine learning and neural networks etc. Please explain in the same paragraph how the AI types propagate the CT or medical image analysis. 

4. Figure 1: Please enlarge the texts in figure one, if needed make it horizontal instead of landscape format. 

5. Discussion: Please add a section on the "strengths and limitations" of the current study. 

Author Response

Q1. I recommend changing the title to this one: " AI-based chest CT analysis for rapid COVID-19 diagnosis and prognosis: a practical tool to flag high-risk patients and lower healthcare costs.

Q2. In the introduction, please highlight the AI-based applications during the COVID-19 period, please write a section on the imaging and analysis part modulated by AI. 

Q3. There are many parts of AI, such as machine learning and neural networks etc. Please explain in the same paragraph how the AI types propagate the CT or medical image analysis. 

Q4. Figure 1: Please enlarge the texts in figure one, if needed make it horizontal instead of landscape format. 

Q5. Discussion: Please add a section on the "strengths and limitations" of the current study. 

Answers:

A1. We agree with your comment and we decided to change the title following your suggestion.

A2. We wrote lines 60-65 in the introduction to answer your comment.

A3. We wrote lines 66-72 in the introduction to answer your comment.

A4. We enlarged the text of Figure 1 and hope it is better now.

A5. We added this section, please report to lines 242-262.

Reviewer 2 Report

In this study, an artificial intelligence system that considers the impact and cost in the planning of hospital resources in the COVID-19 pandemic is proposed. As a subject, it is an interesting and unusual artificial intelligence application, different from ordinary COVID-19 diagnostic prediction studies. The work is well designed and easy to read and understand. The summary is in line with the content and explains the subject well. The following minor changes will make the study better.

Reviews:

1- Line 91: "imaging analysis tools has improved the decision-making process. icolung (icometrix) is artificial intelligence (AI)-based chest CT analysis software for 92 the detection and p..." the first letter of "icolung" needs to be big.
2- The texts in Figure 1 should be enlarged to an easy-to-read level.

Author Response

Q1- Line 91: "imaging analysis tools has improved the decision-making process. icolung (icometrix) is artificial intelligence (AI)-based chest CT analysis software for 92 the detection and p..." the first letter of "icolung" needs to be big.
Q2- The texts in Figure 1 should be enlarged to an easy-to-read level.

Answers:

A1. As the names icolung and icometrix are trademarks, we have to write them in lower case. So, we changed the whole sentence to ensure that this word is not beginning it (see line 66).

A2. We enlarged the text of Figure 1 and hope it is better now.

Reviewer 3 Report

Esposito G and coworkers developed a decision-tree model to evaluate the impact of using an
 artificial intelligence-based chest computed tomography (CT)  to analyze CT scans for the detection and prognosis of COVID-19 cases. The model compared routine practice where patients receiving a chest CT scan were not screened for COVID-19. The model provides a framework for the early detection of COVID-19 cases.
This is a very interesting manuscript. I have only a few points which deserve clarification.

The introduction section is too long. I suggest to short it
The discussion is feeble. I recommend amplifying it. The author should discuss the other method, which gives some information about the prognosis (please cite the following paper PubMed ID 33512742)
In the discussion, the authors can also discuss treatment (please cite the following paper DOI 10.3389/fphar.2020.01124)

Author Response

Q1- The introduction section is too long. I suggest to short it.

Q2- The discussion is feeble. I recommend amplifying it. The author should discuss the other method, which gives some information about the prognosis (please cite the following paper PubMed ID 33512742).

Q3- In the discussion, the authors can also discuss treatment (please cite the following paper DOI 10.3389/fphar.2020.01124)

Answers:

A1. We worked on the introduction to make it 587 words long instead of 910 previously, see lines 32-82 instead of 33-109.

A2. We worked on the discussion, adding a strengths and limitations section (see lines 213-262).

A3. We added a short paragraph on treatment and other methods, which gives some information about the prognosis, as well as the suggested references (see lines 236-241).  

Round 2

Reviewer 1 Report

The revised version of the manuscript has improved a lot and is much easier to follow. I recommend accepting this version.